# Direct Cell-Cell Communication via Membrane Pores, Gap Junction Channels, and Tunneling Nanotubes: Medical Relevance of Mitochondrial Exchange

**DOI:** 10.3390/ijms23116133

**Published:** 2022-05-30

**Authors:** Eliseo Eugenin, Enrico Camporesi, Camillo Peracchia

**Affiliations:** 1Department of Neuroscience, Cell Biology, and Anatomy, University of Texas Medical Branch (UTMB), 105 11th Street, Galveston, TX 77555, USA; 2Department of Surgery and TEAM Health Anesthesia, University of South Florida, 2 Tampa General Circle, Tampa, FL 33606, USA; ecampore@usf.edu; 3Department of Pharmacology and Physiology, School of Medicine and Dentistry, University Rochester, 601 Elmwood Avenue, Rochester, NY 14642, USA; camillo.peracchia@gmail.com

**Keywords:** gap junctions, TNTs, communication, cancer, HIV

## Abstract

The history of direct cell-cell communication has evolved in several small steps. First discovered in the 1930s in invertebrate nervous systems, it was thought at first to be an exception to the “cell theory”, restricted to invertebrates. Surprisingly, however, in the 1950s, electrical cell-cell communication was also reported in vertebrates. Once more, it was thought to be an exception restricted to excitable cells. In contrast, in the mid-1960s, two startling publications proved that virtually all cells freely exchange small neutral and charged molecules. Soon after, cell-cell communication by gap junction channels was reported. While gap junctions are the major means of cell-cell communication, in the early 1980s, evidence surfaced that some cells might also communicate via membrane pores. Questions were raised about the possible artifactual nature of the pores. However, early in this century, we learned that communication via membrane pores exists and plays a major role in medicine, as the structures involved, “tunneling nanotubes”, can rescue diseased cells by directly transferring healthy mitochondria into compromised cells and tissues. On the other hand, pathogens/cancer could also use these communication systems to amplify pathogenesis. Here, we describe the evolution of the discovery of these new communication systems and the potential therapeutic impact on several uncurable diseases.

## 1. The Identification of Membrane Channels

The “cell theory,” proposed by Matthias J. Schleiden and Theodor Schwann in the early 19th century, stated that plant and animal tissue are both made of independent units (later called “cells”) [1,2]; rev. in [3]. This theory implied the existence of a wall-like structure (now known as the plasma membrane) functioning as a barrier to prevent molecules from spreading into neighboring cells and the extracellular compartment or vice versa. However, although this theory did not consider that molecules might move across the plasma membrane, we learned that certain molecules freely cross the plasma membrane soon after publication. Indeed, evidence of the existence of membrane channels was finally reported in the mid-20th century primarily by the studies of Alan L. Hodgkin and coworkers [4,5,6,7,8]; rev. in [9,10].

## 2. Direct Cell-Cell Communication via Gap Junctions

Unequivocal evidence of membrane channels directly mediates cell-to-cell communication in virtually all cells, but circulating cells emerged in the 1960s [11,12,13]. However, knowledge of ionic cell-cell communication in some excitable cells was reported earlier in invertebrate nervous systems [14,15].

In the late 1800s, zoologists discovered that invertebrates such as annelids (earthworms), crustaceans (crayfish), and cephalopods (squid) have large tube-like structures that extend from rostral to caudal ends of the body. Most relevant was the early report of this structure in the squid [16], a structure that would become fundamentally important for studying the ionic basis of the action potential and the function of ion channels in general. Most scientists attributed different functions to these tube-like structures, but Leydig, in 1864, was the first to propose their nervous function [17]. However, his interpretation was ignored for almost a century until John Z. Young eventually proved it right in 1938 [18].

Early in the 20th century, scientists realized that some of these giant structures (axons) were not continuous. The first to report it was George E. Johnson, who discovered discontinuities in giant crustacean axons (*Cambarus* and *Palaemonetes*) [19]. Two years later, these discontinuities were named “septa” in earthworm axons by Howard B. Stough [14]—earthworms have three giant axons: a median and two lateral, all segmented.

In his 1930′s study [15], Stough found that when the median axon is cut, only the anterior part of the worm contracts; in contrast, when just the lateral axons are cut, posterior stimulation causes the posterior portion of the worm to contract up to the cut area. This induced him to conclude that: “*… the median giant fiber conducts antero-posteriorly and the lateral giant fibers conduct postero-anteriorly*”. This, however, was proven wrong by John C. Eccles and coworkers who recorded electrical impulses in isolated nerve cords elicited either in the head or the tail of the earthworm [20]; in their words: “*… the transverse membranes do not influence the conduction of impulses, although the separation of the segments utilizing these transverse membranes appears to be as complete as that existing at vertebrate synaptic junctions*”. Several research teams confirmed their findings a decade later [21,22,23,24]. Stough, nonetheless, deserves credit for demonstrating for the first time the direct cell-cell communication across septal barriers [14,15]. Three decades later, the detailed ultrastructure of earthworm junctions was described by Kiyoshi Hama as a close membrane apposition ~200 Å thick [25].

By noticing the apparent polarization of giant earthworm axons, Stough inadvertently discovered the capacity of injured cells to become independent from healthy cells [15]. This property, now known as cell-cell uncoupling, is mediated by the gating mechanism of gap junction channels. The unpolarized transmission of electrical impulses across septa was also reported in lateral giant axons of crayfish [26] and was later confirmed by intracellular recording [27,28,29,30]. Evidence of electrical cell-cell coupling was reported in other invertebrates, such as lobster cardiac ganglion [27,28,29,30] and muscle fibers [31], and leech segmental ganglia [32,33].

## 3. Direct Electrical Communication between Mammalian Cardiac Fibers

In the early 1950s, Silvio Weidmann made a major discovery while measuring by intracellular recording the myoplasm’s electrical resistance of kid heart’s Purkinje fibers [34]; in his words: “… *The relatively low value of the specific d.c. resistance of myoplasm (twice that of Tyrode solution) suggests (i) that the smaller units making up the Purkinje fiber, the Purkinje cells, are not surrounded by ionic barriers of any importance, (ii) that transverse membranes do not subdivide Purkinje fibers, etc., and (iii) that most of the intracellular ions must be free to move under the influence of an electric field*” [34].

By proving the electrical communication of Purkinje cells, Weidmann’s study provided the earliest example of direct cell-cell communication in vertebrates. In addition, and more importantly, it further proved the “healing-over” ability of these cells, a phenomenon first described by Theodor W. Engelmann [35] and later confirmed by Karl E. Rothschuh [36]. This phenomenon, now called cell-cell uncoupling, is mediated by the gating mechanism of gap junction channels.

## 4. Cell-to-Cell Communication in the Vertebrate Nervous System

Aside from Weidmann’s study on the mammalian heart [34], in the late 1950s, evidence of direct cell-cell communication was only reported in electrically excitable invertebrate cells. Therefore, the 1959′s report on electrical coupling in supra-medullary neurons of a vertebrate made the news [37]. This important discovery proved that electrical coupling is not restricted to excitable invertebrate cells. Soon after, electrical coupling was also demonstrated in two other vertebrate cells: the caliciform synapses of chick’s ciliary ganglion [38,39] and goldfish Mauthner neurons [40].

## 5. Direct Cell-to-Cell Communication in Unexcitable Cells

All examples of cell-cell coupling mentioned above were in electrically excitable cells, so no one expected that unexcitable cells would communicate directly. This was proven wrong in 1964 when two major discoveries demonstrated that all cells, but circulating cells, are coupled. Two independent studies published evidence of direct cell-cell communication in virtually all cells: one on insect glands [11,12] and the other on leech glia [13]. Interestingly, these discoveries were accidental.

Indeed, Yoshinobu Kanno and Werner R. Loewenstein were studying the electrical properties of the nuclear envelope in gland cells of *Drosophila flavorepleta* larvae and were unaware that adjacent gland cells might directly communicate with each other. Much to their surprise, with electrical current injection into the nucleus of one cell, the membrane potential changed not only in the injected cell but also in the neighboring cells [11,12]. Stephen W. Kuffler and David D. Potter also accidentally discovered glial electrical coupling, as their study was trying to determine whether neurons interact with glial cells [13]. These two studies opened an exciting new chapter in the history of cell biology because, after that, no one questioned whether cells were independent units of tissue.

### Gap Junction Structure

Direct cell-cell communication is mediated by channels grouped at cell contact domains known as gap junctions (Figure 1). Each channel is made of two hemichannels (connexons in vertebrates and innexons in invertebrates) that form a hydrophilic pathway spanning two apposed plasma membranes and a narrow extracellular space (gap). In turn, each hemichannel is formed by the radial interaction of six proteins known as connexins in vertebrates and innexins in invertebrates (Figure 1). Connexins/innexins are intramembrane proteins that cross the membrane thickness four times (trans-membrane chains) and contain two extracellular loops and three cytoplasmic domains: a short NH_2_-terminus, a cytoplasmic loop, and a COOH-terminus domain of various lengths (Figure 2). For channels functioning as cell-to-cell pathways well insulated from the extracellular medium, their frameworks (innexons/connexons) must be in register with each other and bind to each other to bridge the extracellular gap (Figure 1).

## 6. Chemical Gating of Gap Junction Channels

Gap junction channels are chemically gated by a rise in cytosolic calcium or hydrogen ion concentrations ([Ca^2+^]_i_ and [H^+^]_i_, respectively). We have reported that cytosolic acidification activates the gating mechanism via a secondary [Ca^2+^]_i_ increase in the high nanomolar to the low micromolar range [41,42]. Over the years, we have proposed a direct role of Ca-calmodulin (Ca-CaM) via a cork-like pore-plugging mechanism, probably involving conformational changes in connexins as well (Figure 3); rev. in [43,44,45,46].

The model proposes two types of CaM-driven gating: “Ca-CaM-Cork” and “CaM-Cork”. In the first, gating involves Ca^2+^-induced CaM-activation. In the second, gating takes place without [Ca^2+^]_i_ rise. The Ca-CaM-Cork gating is only reversed by a return of [Ca^2+^]_i_ to resting values, while the CaM-Cork gating is reversed by trans-junctional voltage (Vj) positive at the gated side [45,47].

## 7. Direct Cell-Cell Communication via Membrane Pores

In the spring of 1968, while we were working on the ultrastructure of gap junctions between lateral giant axons of crayfish located at the ganglia of the ventral nerve cord, we were startled to see areas of cell contact between axons and sheath glial cells in which the two plasma membranes appeared to fuse forming clear images of membrane pores (Figure 4, Figure 5 and Figure 6). Several pores were usually grouped in the same region (Figure 4). In these regions, there were often aggregates of endoplasmic reticulum (ER) tubules, mitochondria, and other membrane structures (Figure 5). The pores were generally 15–20 nm in size (Figure 4), but larger pores were also seen (Figure 5 and Figure 6). Pores of this size are expected to allow wide communication between axoplasm and glial cell cytoplasm. It seemed unlikely that such structures are artifacts of the preparative procedure because these vascular perfused samples’ membranes and other cellular structures were always extremely well preserved (Figure 4, Figure 5 and Figure 6). Indeed, pores between axons and sheath glial cells seemed reasonable from a theoretical viewpoint because only passageways of this size could allow the exchange of molecules such as RNA and proteins as heavy as 200,000 molecular weight (MW) [48,49]. The pores may form transiently but are unlikely to be a preparation artifact because the radius of membrane fusion at the pores (Figure 4, inset “b”) is consistent with the minimum curvature radius of biological membranes.

If indeed pores existed, it would have meant that axons and sheath glial cells could exchange larger molecules than those exchanged by gap junctions. In those years, evidence for the transfer of molecules as large as proteins from glial cells to axons had been reported [50,51]. Still, most of the data supported the transfer from glial cells to axons only, and mechanisms different from cell-cell pores were proposed. We felt that the pores were probably a fixation artifact and decided not to publish these data.

However, *the findings of the Viancour’s team* contacted us a decade later, asking if we had ever seen gap junctions between crayfish axons and their adjacent sheath glial cells. Viancour asked this question because his team had found that when crayfish median giant axons were intracellularly injected with the fluorescent dye Lucifer Yellow CH, the dye rapidly diffused from the axoplasm to the cytoplasm of the adjacent glial cells (Grossfeld, R.M. et al. Soc. Neurosci. Abstr. 6, 385, 1980). The images he sent us were startling as they showed that the nuclei of the glial cells bordering the giant axon were stained by the fluorescent dye (Figure 4, inset “a”). In our answer, we mentioned that while we had reported specialized regions in which axon and glial cell surface membranes were regularly curved and projected into the axoplasm [52], the membranes of these regions of axon-glia apposition were separated by a gap of 130–140 Å, which is much wider than the typical 20 Å gap of gap junctions. Moreover, in freeze-fracture replicas, the structures at the projections did not bridge the extracellular space, so they could not contain cell-cell channels [52], answering the Viancour team’s claim that no gap junction existed between axons and glia.

But, Viancour’s findings prompted us to reconsider our earlier interpretation of the pores and immediately decided to publish the data in Nature [53]. Incidentally, our Nature’s publication, accompanied by our picture on the front page (Figure 7), caused a big uproar. The principal investigator of gap junctions in the nervous system wrote a rebuttal to Nature suggesting the artifactual nature of the pores. However, Nature accepted our answer to his comments and refused to publish the rebuttal in “Matters Arising.”

Soon after our Nature’s publication, Viancour’s team published their data in Nature as well [54]. Indeed, since our original observation of the late 1960s, evidence for the exchange of molecules much larger than gap junction permeants between axons and sheath glial cells of crayfish and squid had surfaced [48,49,55,56,57], indicating that in some tissues, and exceptional circumstances, cell-cell communication could also be mediated by membrane pores. In the four decades that followed our 1981’s publication [53], only one study on crayfish stretch receptors confirmed the presence of membrane pores [58]. Therefore, for several decades the possibility that some cells might be able under special conditions to directly communicate via membrane pores was by and large ignored.

In 2004, however, Rustom and coworkers [59] reported the presence of nanotubular structures between cells. They proposed a novel biological means of cell-to-cell interaction based on membrane continuity and intercellular transfer of organelles. Following this provocative report [59], several publications confirmed that some cells are capable of interacting directly with other cells via tubular membranous structures named “tunneling nanotubes” (TNT), capable of creating large membranous pores by fusing with the plasma membrane of the receiving cell and transferring cytoplasmic organelles and large as mitochondria; rev. in [60,61,62,63,64] among others. Evidence of tunneling nanotubes finally confirmed our data suggesting that some neighboring cells can communicate directly with each other via membrane pores [53].

## 8. TNT or Cytonemes, a New Concept for a Novel Long-Range

### 8.1. Cell-to-Cell Communication System

Both TNTs and cytonemes are thin actin-based long processes that communicate two or more cells. Two different kinds of TNTs or cytonemes have been described as open-ended (a long-range pore that communicates the cytoplasm of 2 or more cells) and a synaptic type of TNT/cytoneme connection like neuronal synapses. The work of Dr. Thomas Kornberg in Drosophila [65,66], as well as the early description of cell-to-cell pores [53], opened a new system of communication that had multiple benefits, including, first, a targeted delivery of a ligand into a targeted cell preventing the diffusion of the ligand into the extracellular space. Second, delivering a high concentration of a factor or receptor at a long distance, and third, establishing a syncytium that can extend up to 500 µm in tissues. Third, a complex transfer of organelles and associated second messengers can change the targeted cell’s metabolism and differentiation. Thus, TNTs/cytoneme/pores provide a new dimension to the cell-to-cell communication that can expand for millimeters depending on the extent of the processes and type of connections. Description of these kinds of cell-to-cell communications has been described in Drosophila wing imaginal disc [65]; abdominal histoblastnest [67], gonadal stem cell niche [68], zebrafish neural plate [69], Chick limb bud [70], and human embryonic stem cells [71]. However, as described below also, several pathogens use these developmental pathways to amplify inflammation and infection.

Early studies were limited to electron microscopy analysis. Several new approaches have confirmed these processes’ essential role in the development of several pathological conditions such as bacterial, viral, and cancer types, as well as acute conditions such as stroke and tissue recovery/resolution. In the developmental examples, most of the Hedgehog (Hh), decapentaplegic (Dpp), wingless (Wg/Wnt), and branchless (Bnl) growth factors are dependent on cytonemes formation and contact with the targeted cell. Preventing these long-range specific signaling compromises development [66,72]. In addition, reevaluation of pioneer data in cell-to-cell communication can be further proven and explained. Thus, a new area or dimension of cell-to-cell communication was born.

### 8.2. Tunneling Nanotubes (TNT)

The development of live-cell imaging and better technologies to fix, identify and characterize the structure and function of these thin processes was essential for obtaining reliable data. TNT have been described relatively recently (early 2000), which may be due to their small diameters and fragility upon fixation. TNTs are long membrane extensions that allow the exchange of several small molecules, vesicles, mitochondria, and pathogenic components, including bacteria, viruses, and pathogenic proteins. TNTs are around 50 to 200 nm in diameter and several µm in length with no contact with the substrate. Works of several laboratories demonstrated that TNT is positive for f-actin, myosin Va, and myosin X, GAP-43, 14-3-3γ, but mostly tubulin negative [60,73,74,75,76]. However, depending on the cell type, differentiation and treatment examined, significant differences in size, diameter, composition (actin and tubulin), transport capacity (transport of molecules), permeability (cut off of the exchanged molecules or components or electrical signals), stability (second, minutes or hours), and regulation has been described [77,78,79].

Several groups have suggested, especially using cell lines, that TNTs are formed from filopodium [59,76,80]. However, our experiments in primary cells indicate that TNT and filopodium are formed in different cellular areas and play different roles (see details below). In conclusion, it is essential to clarify and standardize the terminology or classification of these processes and define a TNT process versus other processes and cell-to-cell pores.

Experiments in several cell lines indicate that TNT has heterogeneous morphology, composition, lengths, and thickness and could form by different mechanisms. TNTs are transient structures with variable lifetimes ranging from a few minutes to hours (see a compressive review [81]. Depending on the TNT analyzed, TNT transfer organelles, ER-related vesicles, endosomal/lysosomal vesicles, mitochondria, and electric and small second messengers, including small proteins, microRNAs, calcium, IP_3_, and several other molecules. The exact formation and transfer mechanism are unknown, but the recent developments in cell-to-cell communication and inter-organelle interactions further developed these research areas [61,82,83,84].

Overall, TNT differs from filopodia in several ways. **First**, TNT structures establish physical contact between connected cells, while filopodia or long processes do not. **Second**, the length of the TNT differs from filopodium but can be similar to the non-connecting process that looks like TNT. Filopodia are 2.0 to 6 µm long, while TNT can reach 150 µm. In contrast, both have diameters between 50 to 200 nm. However, this is upon debate, as will be discussed below. These differences have been described using light and confocal microscopy, but better resolution shows multiple processes per TNT. **Third**, TNT communication requires exchanging cellular or pathogenic components between communicated cells, including second messengers, electrical signals, cytoplasm, vesicles, mitochondria, and pathogenic agents. **Fourth**, these processes are formed from a different cellular area than filopodium in primary cells. Filopodium and other TNT-like structures require interaction with substrates; TNT does not. **Fifth**, identifying at least two different kinds of TNTs, open-ended and synaptic, opens the possibility of having pores at long range with the subsequent consequences of sharing organelles and large molecules.

### 8.3. Major Differences between Gap Junctions, Pores, and TNT

The main differences between these communication systems are the distance between connected cells and the size of the molecules exchanged between connected cells. In the case of gap junction channels and cellular pores, the main limitation of these communication systems is the dilution factor of the cytoplasmic molecules exchanged between communicated cells. Most molecules exchanged by gap junctions/pores are small second messengers up to 1.5 kDa, including cAMP, cGMP, IP_3_, ATP, calcium, glutamate, aspartate, and small peptides and RNA, but no organelles or big protein complexes can be transferred (see [85]). Despite that, diffusion of these second messengers is a major limitation; auto regenerative long-range calcium waves going through gap junctions have been described [86,87]. An example of this long-distance communication system mediated by gap junctions has been observed in the liver but depends on cell-to-cell contact. Hepatocytes expressing vasopressin or glucagon receptors are only localized in the pericentral area of the acini. However, most hepatocytes that respond to vasopressin and glucagon stimulation by degrading glycogen are in the periportal area, at least 100 µm behind the blood flow. Despite this odd localization, upon binding vasopressin or glucagon to their receptors in the pericentral area, retrograde calcium waves are generated and transmitted through gap junctions by an intracellular pathway against the blood flow to reach specialized hepatocytes in the periportal area to release glucose [86,87]. Thus, even though diffusion is a limitation between gap junctions in communicated cells, several systems evolved to amplify these signals to other communicated cells. In the cell-to-cell pores, the possibilities are multiple, including the transfer of cytoplasmic soluble components, organelles, complexes of organelles, and even genetic material due to the large pore size observed by electron microscopy, 130–140 Å [53,58] (Figure 4, Figure 5 and Figure 6).

TNTs, due to a targeted transmission, diffusion of second messengers or pathogens is minimal. Also, to exchange second messengers, TNT allows the exchange of cellular materials, including cellular cargos and second messengers such as vesicles, MCH class I molecules, mitochondria, IP_3_ receptors, and endoplasmic reticulum (ER), calcium and electrical signals [60,88,89,90,91]. Most of the transport through TNT is actin-dependent because blocking actin reduces the transfer of these cellular components [59,92].

Even though TNT is a novel discovery, several TNT-like structures have been previously described in neuronal cells, migratory cells, pathfinding processes, and invadopodia in the case of tumor cells. Also, re-examinations of older reports show TNT-like structures in several inflammatory conditions. For example, in in vitro pathological conditions, the formation of TNT-like structures has been observed after infection with *Listeria monocytogenes* and *Mycobacterium bovis* [93,94,95]; in astrocytes treated with H_2_O_2_ [96], microglia activated with PMA and calcium ionophore [97], monocyte/macrophages treated with LPS plus IFN-γ [98], mouse neuronal CAD cells infected with exogenous PrP [99] and more recently in lymphocytes infected with HIV [91] and in human macrophages infected with HIV (Figure 8) [79]. In vivo, TNT-like structures have been observed in *Drosophila* [100,101], between immune cells in lymph nodes (see review by [89,102]), in the dendritic cells (DC) of the gut [103,104], and the MHC class II^+^ cells in the mouse cornea [105]. The clinical relevance of TNT has already been demonstrated in several pathological conditions (see below). As an example, Mesenchymal Stromal Cells (MSC) have been shown to improve the clearance of bacteria in models of acute respiratory distress syndrome (ARDS) by mitochondrial transfer from MSC to macrophages via tunneling nanotubes [63]. However, all new and old reports require the classification of these processes.

### 8.4. TNT-like Structures In Vivo

Currently, most of the data available about TNT have been obtained in vitro. However, whether these processes are present in vivo is still debated, mainly due to the unclear definition and lack of makers to identify TNTs. However, a consensus point for several laboratories is that TNT requires communication with another cell and exchange signaling, cargo, or organelles components. So far, in vivo, these two components have not been proven.

Currently, several suggestive publications show TNT-like structures in MHC class II dendritic cells in the corneal stroma [105], non-neural ectoderm cells in the midbrain [106], trophectoderm cells in the neural crest [107], epiblast cells in the blastula [108] and the ectoderm [109]. Also, several groups have suggested that neuronal processes may be similar to TNT. However, we will exclude from this review this possibility because dendrites are a clear formation, transport, and stability and lack many components of TNT.

One of the best examples of TNT in vivo has been described in flies. Only recently, the role of TNT or cytoneme has been well described for several groups, indicating the critical role of these processes in the signaling and delivery of developmental proteins such as Dpp, fibroblast growth factor (FGF), Hedgehog, Wingless (in flies), sonic Hedgehold (chick limbs), and wnt proteins (in developing zebrafish) directly into sites of cell to cell contact without dilution of these factors into the extracellular space [67,68,110,111,112,113]. TNT is a changing paradigm in the area because it was assumed that most of these proteins were released into the extracellular space. Roy et al. [114] describe that Dpp and FGF signaling is required for signal-producing wing disc cells, and several of the genes activated by these factors are required for TNT (cytonemes) [114,115] or synapses formation [114,115,116]. Most of the evidence seems to indicate that other factors important in the development of flies are TNT-mediated, such as fibroblast growth factor branchless [117], Dpp [113,114], EGF, Hh [67], SHh [70], and Notch [114,115,116,118]. Most of these factors are also present in humans, but the role of TNTs is unknown. Thus, it may be possible to extrapolate some of these data into human development.

In chicken embryos, the filopodia-like protrusion (or TNT-like processes) span the sub-ectodermal space to establish contact with the ectoderm. These actin and tubulin positive processes require Rac1 to form and participate in the retrograde trafficking of the transmembrane Wnt-receptor Frizzled-7. Another great example of TNT-like processes has been described in zebrafish embryos; several manuscripts describe their key role in controlling the spread of Wnt morphogens such as wnt8a that activates Wnt/β-catenin signaling pathways upon TNT contact. The formation and transfer of these molecules were cdc42/N-wasp dependent [119]. However, experiments in chicks suggest that shh-positive transfer and spread are dependent on long TNT-like processes (hundreds of micrometers) that are cdc42 negative [70]. Thus, in these experiments, using actin blockers such as latrunculin B, cytochalasin D, or overexpression of IRSp53 reduces filopodium formation, suggesting that actin assembling and f-actin-bundling are necessary for filopodium formation [69]. In these experiments, it has been suggested that the formation of filopodia (no-TNT) can be initiated in response to morphogenic factors such as FGF by the mDia3C mechanism [120]. Kondo’s laboratory (Osaka University, Japan) is a pioneer in identifying TNT-like structures in controlling zebrafish pigmentation [121,122]. This transfer of information between xanthophores extends processes (or dendrites) to melanophores, and contact with these processes induces the melanophores to migrate away, changing the pigmentation pattern [123].

In heart development, TNT-like processes participate in the cardiac differentiation of progenitor cells that give rise to the heart tube. In this case, filopodial extensions from the dorsal pericardial wall are directed into the endoderm and adjacent mesenchymal cells to coordinate signaling events related to the hedgehog, Notch, BMT, and non-canonical wnt pathways and to maintain cardiac progenitor status during heart tube extension [124]. Genetic deletion of tbx (22q11.2 deletion syndrome gene) and aPKCζ activator resulted in the loss of filopodia and decreased FGF signaling, reduced proliferation, and ectopic differentiation [124,125].

### 8.5. TNT Expression under Pathological Conditions

The data discussed above clearly indicate that TNT-like processes are expressed to deliver and spread in a specific manner wnt and notch-related morphogenic molecules during development. Thus, are pathogens that induce TNT for spread and inflammation using these mechanisms?

Interestingly, viruses, such as African swine fever, Ebola, Herpes Simplex, Marburg filoviruses, SARS-CoV-2, and Poxvirus Vaccinia encode viral factors or alter cell activation to induce the formation of filopodia structures to allow viral trafficking between the extracellular matrix or environment into cells [126,127,128,129,130,131,132], suggesting that viruses are adapted to use filopodia and TNT-like structures to improve viral spread.

Cytonemes (also called TNT, membrane tubulovesicular extensions, or TNT-like structures) have been described in neutrophils directly contacting yeast or bacteria. The role of TNT is to provide a directed delivery of anti-pathogen molecules at the site of infection. This group using human neutrophils describes that the mechanism of formation of these processes is GTPase dynamin or actin-dependent [133]. However, most of these reports lack an accurate characterization of these processes. These data indicate that pores and/or TNT can be present in activated immune cells.

### 8.6. Molecular Mechanism of TNT Formation

Despite the multiple questions in the TNT area, several groups using cell lines of different origins have described increased TNT formation in several stress conditions. Still, how they are formed and the signal(s) involved in directing formation and stability is unknown. Several reviews and primary communications indicate that M-sec, part of sec6 family proteins, and p53 activation are the key master regulator of TNT formation in several cell lines [134,135]. The mechanisms by which M-sec activates Ras-like small GTPase ral-A promote the binding to filamin. This protein cross-links actin filaments and promotes filopodia formation, as well as cdc42 [134,135]. Also, H_2_O_2_ treatment or serum starvation in neuronal cell lines is dependent on p53 and subsequent activation of epidermal growth factor (EGF) receptor, Akt1, PI3K, and mTOR [136]. Despite the beauty of this signaling system in cell lines, our data in primary human cells are confusing about the role of these proteins in TNT formation. Clearly, there is a missing link between primary cells and immortalized cell lines, where most of them have affected p53 expression and signaling.

Our data, using human primary macrophages or T cells, indicate that blocking M-sec expression using siRNA or treatment of primary cells with p53 activators such as CGK733, p53-stabilizing agents such as CP31398, reactivators of P53 mutant such as NSC319726, or MIRO-1, a proapoptotic p53 related protein, does not alter the formation of TNT in basal or in HIV-infected conditions or the form of TNT including fused open-ended TNT or synaptic kind of TNTs. Interestingly, several TNTs behave like pores (described above—direct cell to cell communication), though some were also sensitive to gap junction blockers suggesting that the synaptic kind of TNTs contain gap junctions (GJ). In addition, in different systems, functional GJ channels increase the invasion and dissemination of *Shigella* in epithelial cells [137] and increase toxicity between connected HIV-infected cells [138,139,140]. Suggesting the possibility that particular infectious agents, such as *Shigella* and HIV, can use GJ channels to sensitize uninfected cells and spread infection/toxicity to healthy cells, but whether TNT or TNT-like structures play a role in these infectious diseases is still under investigation. Alternatively, GJ benefits the host immune system by mediating a cross-antigen presentation phenomenon. This enables coupled cells to share viral peptides (antigens) and trigger a response in CTL cells, even when some cells were never directly exposed to the pathogen [141]. GJ-mediated immune coupling suggests the possibility that GJ expressed by monocytes/macrophages in inflammatory conditions cross-present antigens to lymphocytes and other inflammatory cells to maintain an immune memory in cells never exposed directly to specific antigen [141]. In the agreement, Cx43 is recruited to the immunological synapses during T cell priming, suggesting that GJ and hemichannels also participate in antigen presentation [142]. Thus, some pathogens such as HIV may potentially use GJ and maybe TNT (see below) to spread toxicity and its normal role in enhancing the immune response. However, most viral and bacterial infections down-regulate expression and GJ function, probably to reduce damage. Thus, pathogens require hijacking this communication system to spread toxicity and damage. We propose at least three kinds of direct cell to cell communication: GJ, pore mediated (close interactions), and TNT mediated (pore and synaptic type, long-range combination of both communication systems).

Several pioneer studies using T cells described the potential role of TNT in the propagation of Fas ligand-related cell death signals between communicated T lymphocytes [143,144]. Also, TNT has been proposed to provide an exchange pathway between injured cardiomyoblasts or endothelial cells by mesenchymal stem cells (MSC) through transferred mitochondria [145,146]. The direct transfer of mitochondria between connected cells, pores, or TNT provides metabolic advantages for metabolic distress cells, such as during stroke [147]. However, tumor cells utilize mitochondrial transfer in several types of cancers to adapt the tissue to the metabolic tumor environment [61,148,149,150,151,152]. However, whether these cells also show direct pore communication is unknown.

### 8.7. Cell-Cell Mitochondrial Transfer via Tunneling Nanotubes

Overall, TNTs are minimally expressed under physiological conditions [84,153]. However, they proliferate under stress and in response to several pathogens such as HIV, as well as pathogenic conditions such as cancer, Alzheimer’s, Parkinson’s, and other neurodegenerative diseases. The proposed function of TNT in pathogenic conditions is to amplify toxicity, apoptosis, and pathogenesis between connected cells. TNTs are generated in the “sick” cells into healthy surrounding cells to maximize efficiency in these conditions. In contrast, TNTs are also induced by stress conditions, including stroke, spinal cord, and traumatic brain injury. In these cases, TNTs are generated from compromised cells (cells in the penumbra area) into healthy areas. In these cases, it has been proposed that compromised cells “request” healthy factors and organelles to survive the damage in these conditions. These two mechanisms of TNT-induced formation, pathogenic and recovery, could be therapeutically targeted to regulate TNT formation and associated transport. *In addition, several groups described exosomes containing mitochondria elements* [154,155]*. However, we will not discuss this type of mitochondrial element transfer for two reasons; first, released exosomes diffuse into the extracellular space losing the advantage provided by GJ, pores, and TNTs by maintaining a close mitochondrial transfer. This is essential to transfer enough organelles or their DNA to change the metabolism of the targeted cell. Second, there is a lack of proper cell-to-cell selectivity. This despite that exosomes had potential receptors to provide selectivity. Even though the exosomal mediated mitochondrial transfer or some of its elements can be exciting, whether the exosomal transfer can effectively change the phenotype of the targeted cell is still unclear.*

Several groups suggested that TNT-transmitted organelles such as lysosomes and mitochondria could alter the fate of the recipient cells. Mitochondria are interesting TNT-transmitted organelles. In addition to their traditional role as cell powerhouses, mitochondria also play a key role in calcium buffering, ER interactions, ROS generation, apoptosis, ER stress response activation, and a broad variety of mitochondrial dysfunction. Furthermore, mitochondrial genetic alterations have been associated with metabolic diseases, including Alzheimer’s disease (AD) and Parkinson’s disease (PD), muscular dystrophy, and the process of normal aging [156]. Thus, TNT exchange of compromised mitochondria could result in accelerated disease.

The mitochondrial TNT-mediated exchange has been proposed as a general principle to rescue damaged cells and genetic diseases [157,158,159,160]. Mitochondrial transfer can rescue aerobic respiration in stem cells [161], mitochondrial function even if mtDNA is damaged [162], and epithelial function if mitochondria are transferred from stem cells [163]. However, few studies demonstrated that TNT could be the mechanism of mitochondrial transfer and, more importantly, the transferred mitochondria could change the TNT-targeted cell’s metabolism or function. For example, microinjection of intact mitochondria into oocytes prevented apoptosis [164]. These data indicate that these communication systems can change the metabolism of the targeted cell, probably due to the highly concentrated organelle transfer in a cell-to-cell specific manner.

Our data demonstrated that TNT generated in response to heterogeneous interaction between tumor cells and stromal cells promotes the transfer of a unique type of mitochondria from tumor cells into surrounding astrocytes resulting in their metabolic and hypoxic adaptation [61]. Currently, it is unknown whether the TNT-transferred mitochondria work independently in the targeted astrocytes, fused with the existing ones, or replace the mitochondrial system of non-cancer cells. However, blocking TNT mitochondrial transfer prevents healthy astrocytes’ adaptation to tumor-related conditions such as ischemia. More concerning is that radiation alone could induce TNT formation and transfer, suggesting that anti-cancer treatment may contribute to the tumor’s adaptation to subsequent treatment or tumor survival. In contrast, UV treatment induces the formation of TNT to enable the transfer of healthy mitochondria from healthy cells to rescue cell apoptosis with compromised DNA [165]. Both conditions indicate that DNA damage or compromise activates TNT formation and associated mitochondrial transfer.

Our data indicate that mitochondrial transfer enables “healthy cells,” such as primary astrocytes, to become tumor-like cells from a metabolic point of view. Furthermore, we identified a critical “fingerprint” of the TNT-transferred mitochondria in glioblastoma. Mitochondria expressed few genes and depended on nuclear transcription for the synthesis of all proteins for the organelle; however, we identified critical mutations in the mitochondrial DNA that favor the use of glutamine/glutamate instead of glucose and lipids to produce energy, a critical phenotype on aggressive tumors [166,167]. Glutamine use is a marker of aggressive glioblastoma [168]. Glutamine is a nitrogen source to synthesize nucleotides, amino acids, and ATP and is a major anaplerotic precursor for the TCA cycle. Glutamine dependency in different cancers promotes invasion and tumor aggressiveness [169]. Also, glutamine dependency can downregulate glycolysis reducing glucose uptake and lactate production, contributing to the Warburg effects observed in ischemic tumors. We identified that TNT formation also protects astrocytes from hypoxic conditions. Tumor persistence and growth relay in the survival of cancer stem-like cells promoted by a hypoxic microenvironment [170,171,172]. It is believed that low oxygen levels prevent ROS formation and DNA damage, nuclear and mitochondrial. Our data support both ideas that mitochondrial transfer changes the metabolism and contributes to adaptation to hypoxic conditions—both essential components to promote tumor growth.

Interestingly, TNTs are the only communication system that enables targeted and selected transfer of pathogenic material, including mitochondria. Thus, blocking TNT formation and associated communication could provide additional treatment to reduce tumor adaptation and prevent chemotherapeutic resistance [157,173,174,175,176,177]. Also, because TNTs are not expressed in healthy adult tissues, the expected toxic effects may be reduced.

We recently demonstrated TNT communication between heterogeneous glioblastoma tumor cells based on their resistance to TMZ and radiation therapy. First, surprisingly we identify that anti-tumor treatment promotes TNT formation and transport. Second, TNT concentrates the anti-apoptotic enzyme MGMT and distributes it among cells with insufficient MGMT expression to avoid apoptosis in response to TMZ and radiation treatment. Blocking TNTs prevented MGMT diffusion and survival of cells with minimal MGMT expression. Third, we identify that tumor TNTs are selective because MGMT protein, but not its mRNA, was transferred in vitro and in vivo in glioblastoma and breast cancer.

In agreement with the data in glioblastoma, in HIV-infected conditions, the mitochondrial transfer is essential to spread infection and associated inflammation. In immune cells, it has been demonstrated that TNT can propagate cell death induced by Fas ligand among T lymphocytes [143,144]; however, how amplification of disease and their protective mechanism remain unsolved. Our data on HIV indicate that viral spread is associated with mitochondrial function, not only to provide a formidable amount of ATP to induce the TNT formation and associated transfer but also because some of these organelles nurse some of the infectious HIV material transferred by TNTs. However, the contribution of these organelles is still unknown.

Several manuscripts identified specific molecules involved in mitochondrial transport, such as the protein Miro. Miro 1 and 2 are mitochondrial outer membrane-associated GTPases that regulate mitochondrial trafficking and distribution by coordinating actin- and microtubule-dependent mitochondrial movement due to interactions with TRAK and KIF5 [178]. It was reported that Miro1 knockout significantly inhibited microtubule-dependent mitochondrial motion, whereas Miro2 deficiency showed no effect on microtubule-dependent mitochondrial movement, suggesting that Miro1 is important for microtubule-dependent mitochondrial trafficking [179].

Several stimuli of TNT formation, such as the Fas-ligand receptor in the immune system or M-sec, a protein associated with the component Sec6 of the exocyst complex required for the docking of exocytic vesicles on the plasma membrane [143], have been discovered. Involved molecules in the TNT formation machinery are Rho and Ras small GTPases families such as Cdc42, Ral or the exocyst effector, myosin V/X, and the transmembrane MHC class III protein LST1. Also, Dephosphorylated β-Ca^2+^-Calmodulin-dependent protein kinase II (βCaMKII) has been shown to stabilize TNT formation [180]. βCaMKII not only stabilizes F-actin and binds to G-actin, preventing its nucleation, and in TNT, the protein is localized at the base of the TNT. These findings suggest that the TNT formation might be like filopodia and lamellipodia regulation mechanisms.

First, the interaction of M-Sec with RalA is necessary for the formation of TNT. Further, TNT formation has been associated with M-Sec expression. Also, RalA has been shown to interact with filamin to promote filopodia formation; however, whether these interactions play a role in TNT formation, stability, and associated transport is unknown.

Overall, identifying TNT as a new mechanism of long-range communication during development and pathological conditions opens several potential innovative therapeutic possibilities. First, TNTs are minimally expressed in healthy individuals; Second, TNTs proliferate upon pathogenesis; Third, pathogens such as viruses and bacteria use them to amplify inflammation and infection; and Fourth, traumatic conditions use TNT to rescue compromised cells. Thus, identifying potential blockers or inducers of TNT and the cargo associated with its formation can provide novel medicine and therapeutic approaches for life-threatening conditions.

### 8.8. Clinical Relevance of Tunneling Nanotubes

Currently, there are few examples of TNT in vivo, including in HIV animal models [77] and glioblastoma tumors [61,82,181]. However, observation of TNT processes in clinical samples has been proposed but not examined [182,183], though several laboratories are working on similar microtentacle (well-defined tubulin content) processes. Microtentacles are microtubule-based protrusions with high actin and vimentin content, as originally observed in detached breast cancer cells [184]. The main proposal is that detached cancer cells may influence the metastatic and chemoresistance profile of metastatic cells. Both these are also critical functions of TNT. However, a critical difference with TNT is that they do not communicate with cells until they establish contact with endothelial cells or tissues to invade. Thus, these microtentacles could establish TNT-mediated communication with other cells at one point. The growth of the microtentacles is regulated by kinesins and microtubule-associated proteins, including tau [185]. Also, cdc42 acts as molecular switches by alternating their active GTP-bound form and inactive GDP-bound form in cells. Aberrant cdc42 activity is implicated in tumor metastasis by promoting transendothelial migration [185,186]. Currently, several clinical trials block and/or promote microtentacle extension or attachment [185,187,188,189]; however, whether these drugs alter TNT formation or associated communication is unknown. 

Interestingly, tau induces the formation of microtentacles and reduces the effects or binding of the anti-cancer drug paclitaxel [190]. However, these results had significant implications for cancer, Alzheimer’s disease, and other tau-related pathologies [191,192,193,194].

The identification of the mechanism of TNT formation (as well as other types of long communication processes) and associated function in cancer and other pathologies is an essential step to move forward in the TNT field from a mostly in vitro system into in vivo and the use of potential drugs to block or promote TNT formation. Block to prevent the spread of pathogenesis (cancer, Alzheimer’s disease, and other related diseases). In this case, TNT formation spread toxic organelles and second messengers into neighboring healthy cells. However, it also promotes TNT formation to rescue damaged cells from ischemic events such as stroke and traumatic events. In this case, healthy cells send healthy mitochondria and second messengers to compromised cells. These mechanisms denote a high impact of TNT formation and associated transport on human diseases that have not been exploited until now. In addition, there is no formation of TNTs in healthy individuals, increasing the possibility of not having side effects. Thus, the clinical and translational possibilities are highly significant for several incurable diseases and provide an alternative treatment for reducing brain/spinal cord physical damage or preventing/recovering ischemic events such as heart attacks and stroke.

A recent commentary [64] updated the present trends in mitochondrial research to three areas: (1) the causal role of mitochondrial dysfunction in several neurodegenerative ailments, such as Parkinson’s and Alzheimer’s disease; (2) the control of Ca^2+^ milIeu in selected intracellular micro-domains, changing intracellular responses; and (3) microvesicle (mV) or TNT-mediated transfer of mitochondria within different cells and tissues, of healthy or of degenerated mitochondria, as a mechanism of functional renewal in diseased cells, or fostering propagation of cell distress or pathology, such as in metastatic cancer evolution.

Until recently, mitochondria were considered to remain housed intracellularly for the life duration of each cell. The current review builds on the evidence that the intercellular exchange of mitochondria (and other intracellular organelles) continues during normal tissue growth and evolution. This exchange between cells appears robust during normal development. It continues in various pathological conditions, as demonstrated separately, especially in the lungs, kidneys, and the myocardium, especially in the ischemic brain’s repair. The earliest paper [195] demonstrated that bone marrow-derived mesenchymal stem cells (MSC) reduce acute lung injury in animals challenged by bacterial infection. They also suggested that MSC has a crucial role in preventing VILI (volume-induced lung injury) in healthy rats subjected to high volume ventilation. A potential therapeutic tool for patients emerged from this early paper, suggesting several clinical trials. In a later publication, Chen et al. [196] summarized in a table the several clinical application of mitochondrial transplantation by detailing a register of ongoing human clinical trials. This paper also discusses how mitochondrial transfer to damaged cells can help revive cells energetics in recipient cells.

Jackson et al. [63] demonstrated that mesenchymal stromal cell MSCs exert their antimicrobial effect via macrophage phagocytic activity, which can be mediated through mitochondrial transfer. In this model, fluorescent imaging and flow cytometry demonstrated extensive mitochondrial transfer from MSC to macrophages, which occurred at least partially through TNT structures. Finally, in a different opinion paper and a more recent review [197,198], we can appreciate the effect of mitochondrial transfer as a therapeutic strategy against ischemic stroke. These comments discuss how healthy mitochondria can be transferred to stroke cells during ischemic stroke.

Lippert and Borlongan [199] extend these concepts and elucidate the prophylactic use of hyperbaric oxygen treatment. This novel paper describes the mechanism of action of preventive treatments by hyperbaric oxygen exposure. For the first time, it demonstrates how a treatment preceding the exposure to a noxa could transfer the resistance mechanism to a tissue. Since oxygen levels inside a tissue are fleeting and can only last for a short time, the authors visualized directly mitochondrial transfer between cells induced by hyperbaric oxygen pretreatment. The process results in the transfer of healthy mitochondria and induces increased resistance of the receiving cells to the inflammatory response typical of ischemia and traumatic brain injury. These results are robust, with preconditioning doubling the salvage rate of the cells subjected to the experimental trauma, and explain the long-term effect of this technique which is clinically used in several surgical and medical patients.

A comprehensive review of mesenchymal stem cells and their ability to provide mitochondrial transfer has been organized by Cheng et al., 2019 [200]. In this review, the authors underline the improvement in clinical outcomes obtained in animal models of alveolar cells and pulmonary infection in improving the function of cardiomyocytes and the transfer of mitochondria via astrocyte into ischemic brain cells. However, it also demonstrates that cancer cells may enhance their ability to produce metastases by shifting damaged mitochondria and inducing chemoresistance. Because of this, the title of this review contains the definition: of a double-edged sword. The highlight of his article is a summary of different mechanisms of mitochondrial transfer via TNT and cell fusion or through GJ. The transfer mechanism is reviewed in this paper. It illustrates the formation of TNT between the recipient cell and the donor cell, or the organization of GJ, via the help of the connexin-43 or direct transfer of mitochondrial DNA from MSC cells via extracellular vesicles (eV), and also the possibility of selective loss of donor nuclei after complete cell fusion. They mediate the transfer of chemoresistance in a tumor and can be utilized as a clinical tool to provide pharmaceutical blockage sites and restoration of healthy tissue.

A recent review by Crew et al. [201] underlines how inter-organ transport of Mitochondria can be generated from the abundant supply provided by adipocytes. They can readily donate eV-encapsulated mitochondria from oxidatively damaged adipocytes. These are injected into the circulation and can be uptaken in cardiac tissue. This up-regulation of cardiac tissue results in resistance to cardiac ischemia, a paradoxical observation in obese patients, repeatedly reported.

Finally, in a recent review article Liu et al. [202] provide specific tables summarizing intercellular mitochondrial transfer in physiological conditions between cells and between different tissues and an extensive list of dozens of papers providing transfer in animal models and the various tissues studied so far. This comprehensive review also discusses all the mechanisms of transfer and especially the stimuli necessary to evoke the beginning of the intercellular transfer. They learn the means of organization and cell connection to propagate and initiate intercellular communication and transfer of organelles between cells. The evolution of this novel field will provide clinical novelty for maintaining organs suffering from aging and ischemic cells. And hopefully offer a new mechanism to support chemotherapy and limit cancer expansion.

Therefore, mitochondrial transfer is a frequent phenomenon associated with various bodily physiological and pathological activities. It may result in the restoration of normal physiological functions, and may help in the recovery from disease in animal models, as it already has in a few patient cases. Meanwhile, the mechanism of inhibiting transfer from stromal cells to cancer may develop the potential therapeutic target of a new anti-tumoral agent.

## Figures and Tables

**Figure 1 ijms-23-06133-f001:**
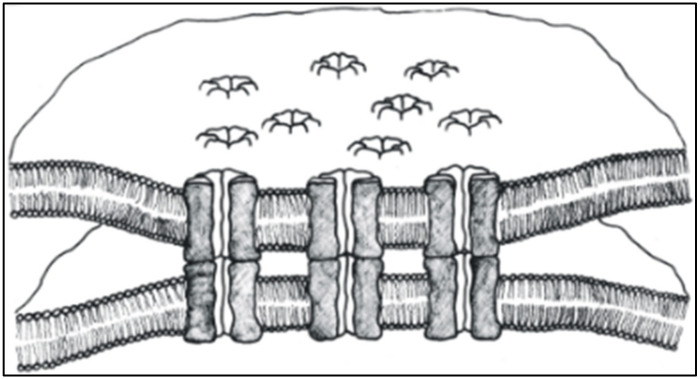
Gap junction model. Each channel comprises two hemichannels that form a hydrophilic pathway spanning two apposed plasma membranes and a narrow extracellular space (gap). Each hemichannel is formed by the radial interaction of six proteins (connexins/innexins).

**Figure 2 ijms-23-06133-f002:**
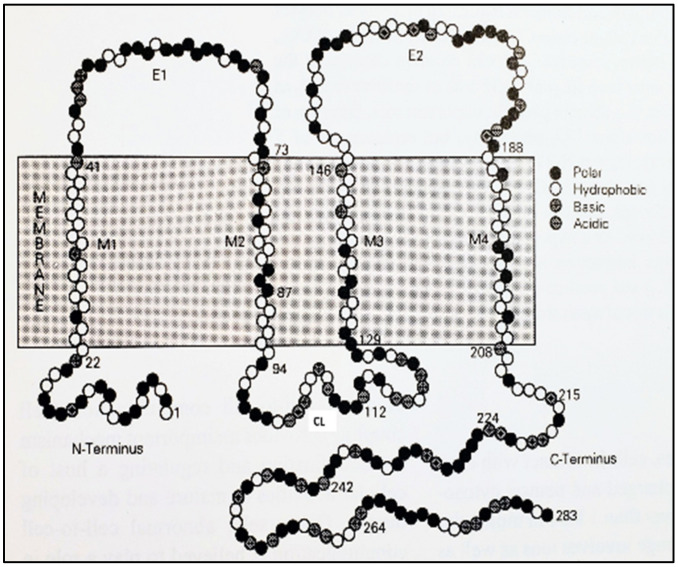
Model of connexin 32 (Cx32) topology. The connexin spans the bilayer four times (M1–M4) and has both NH_2_- and COOH-termini at the cytoplasmic side of the membrane, forming two extracellular loops (E1, E2) and one cytoplasmic loop (CL).

**Figure 3 ijms-23-06133-f003:**
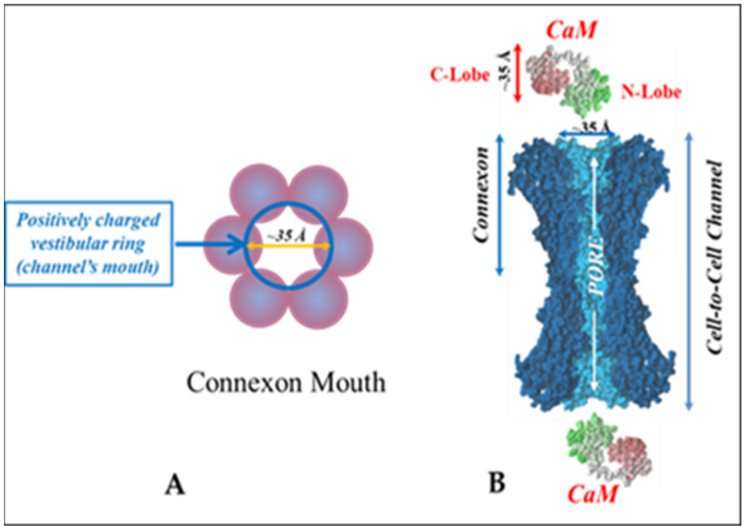
Cork-gating model. Both the positively charged channel’s mouth and the negatively charged CaM lobes are ~35 Å in diameter (**A**). Therefore, a CaM lobe could fit well in the channel’s mouth (vestibule) and electrostatically interact with it. In (**B**), the channel is split along its length to display the pore diameter (light blue area) along the channel’s length. Both CaM and connexon images (**B**) were provided by Dr. Francesco Zonta (Venetian Institute of Molecular Medicine, VIMM, University of Padua, Italy).

**Figure 4 ijms-23-06133-f004:**
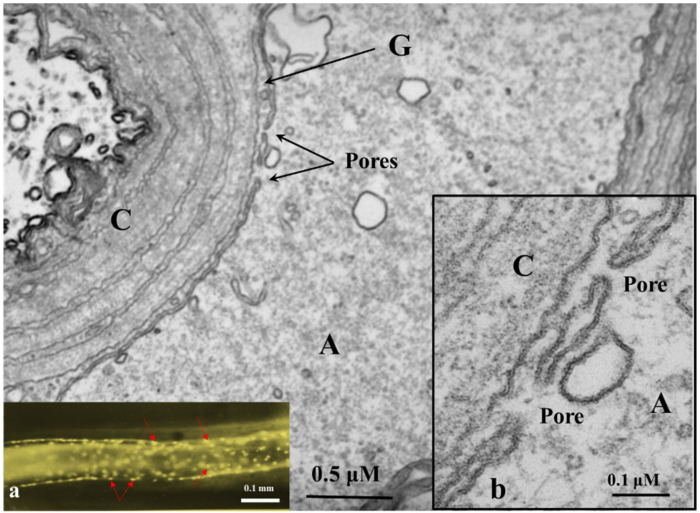
Thin section of an axon of a crayfish abdominal ganglion. Axonal and glial cell plasma membranes fuse, forming two pores (P). Inset “a” shows a fluorescence micrograph of a median giant axon intracellularly injected with Lucifer Yellow CH; note that the nuclei of the adaxonal (sheath) glial cells are stained by the fluorescent dye (Inset “a,” red arrows). Inset “b” is an enlargement of the pore region. A, axon; G, glial cell; C, Adjacent glial connective tissue sheath, non-cellular. The fluorescence micrograph of inset “a” (provided by Dr. Terry A. Viancour) represents a crayfish median giant axon. The electron micrograph is an original, unpublished image taken by one of the authors (C. Peracchia).

**Figure 5 ijms-23-06133-f005:**
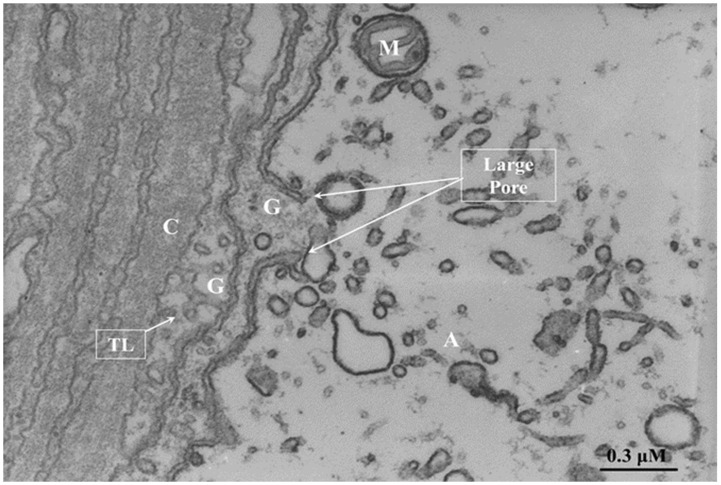
Thin section of an axon of a crayfish abdominal ganglion. The arrows point to a large pore area. Note the accumulation of cytoplasmic organelles near the pore region. A, axon; G, *glial satellite cell*; C, *Adjacent Glial connective tissue sheath, non-cellular*; M, mitochondrion; TL, tubular lattice (Peracchia and Robertson, J. Cell Biol. 51, 223–239, 1971). The electron micrograph is an original, unpublished image taken by one of the authors (C. Peracchia).

**Figure 6 ijms-23-06133-f006:**
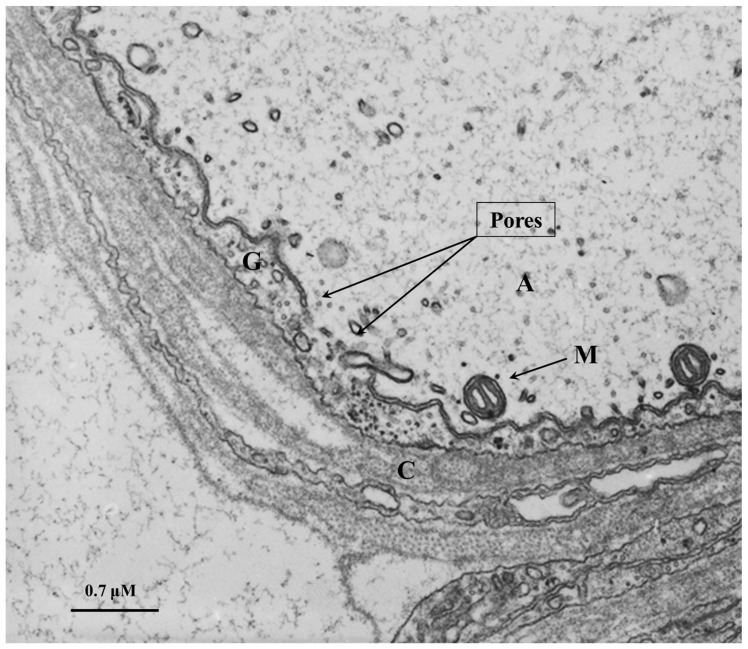
Thin section of an axon of a crayfish abdominal ganglion. The arrows point to a large pore area. A, axon; G, glial cell; C, Adjacent Glial connective tissue sheath, non-cellular; M, mitochondrion. The electron micrograph is an original unpublish image taken by one of the authors (C. Peracchia).

**Figure 7 ijms-23-06133-f007:**
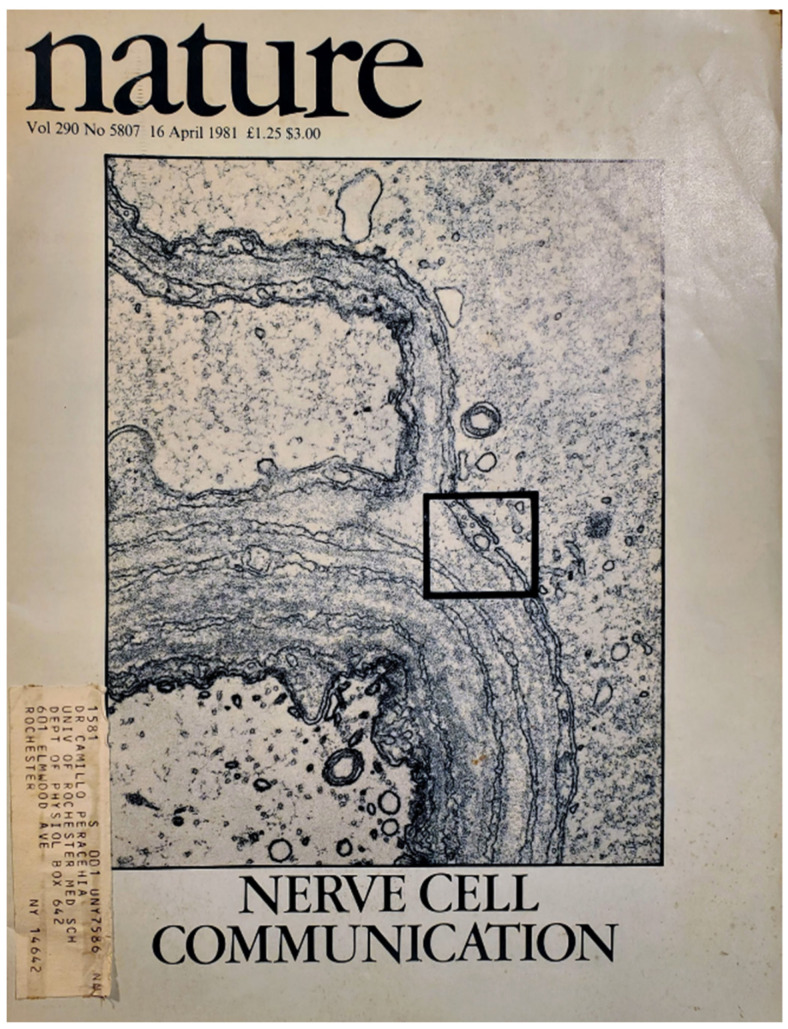
Nature’s front page shows one of our pore images.

**Figure 8 ijms-23-06133-f008:**
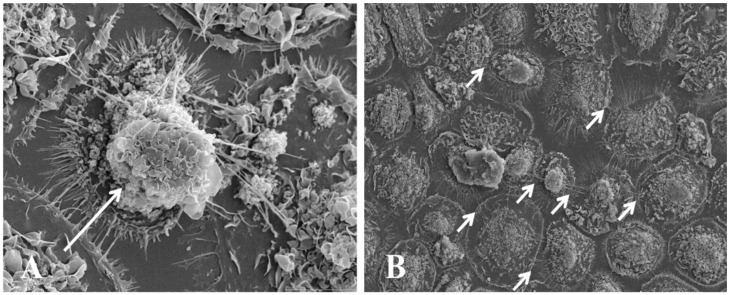
Morphological characteristic of TNT in human macrophages. Representative image after 3 days post HIV infection analyzed by SEM. (**A**) Representative image of HIV-infected cultures after 3 days post-infection. (**B**) High magnification of the end of the TNT process in HIV-infected conditions. Arrows denote TNT. *n*  =  5.

## Data Availability

No applicable.

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
