# Peer review of "Direct Cell-Cell Communication via Membrane Pores, Gap Junction Channels, and Tunneling Nanotubes: Medical Relevance of Mitochondrial Exchange"

_ijms, 2022, doi:10.3390/ijms23116133_

Round 1

Reviewer 1 Report

The manuscript entitled ‘DIRECT CELL-CELL COMMUNICATION VIA MEMBRANE 2 PORES, GAP JUNCTION CHANNELS AND TUNNELING 3 NANOTUBES:MEDICAL RELEVANCE OF MITOCHON- 4 DRIAL EXCHANGE. described the exchange of mitochondria in different ways and explained their clinical applications. the data is exciting and valuable however a schematic figure may be helpful to readers. The author may discuss the possibility of exosome-based mitochondria shuttling. 

Author Response

May 23, 2022

To Dr. Megan Dong

Enclosed is our revised manuscript entitled "Direct cell-cell communication via membrane pores, gap junction channels and tunneling nanotubes: medical relevance of mitochondrial exchange " Manuscript number, IJMS-1729888. As requested by the reviewers and the Editors, we answered all the concerns and comments indicated by the reviewers. Thank you for the constructive comments and suggestions that certainly strengthened our manuscript. We included a point-by-point response to both reviewers, and the changes in the manuscript are noted in italics letters. We hope that our manuscript is now acceptable for publication in IJMS.

Reviewer #1.

Reviewer #1:Comments and Suggestions for Authors

The manuscript entitled ‘DIRECT CELL-CELL COMMUNICATION VIA MEMBRANE PORES, GAP JUNCTION CHANNELS AND TUNNELING NANOTUBES: MEDICAL RELEVANCE OF MITOCHON- DRIAL EXCHANGE. described the exchange of mitochondria in different ways and explained their clinical applications. the data is exciting and valuable however, a schematic figure may be helpful to readers. The author may discuss the possibility of exosome-based mitochondria shuttling. 

Answer:  Thank you, and we included all your suggestions.

Thank you for all your constructive comments and suggestions. We believe our findings are extremely novel and exciting and would appreciate their consideration for publication in IJMS. Changes in the text are indicated by red Italics letters.

Sincerely,

Eliseo Eugenin. Ph.D.

William D. Willis, Jr., M.D., Ph.D. Professorship in Neuroscience 

University of Texas Medical Branch (UTMB)

Department of Neuroscience, Cell Biology, and Anatomy 

105 11th Street 

Galveston, Texas, 77555

Office: (409) 772-7705

Lab    :(409) 772-0155

Cell Phone: (845) 490-3332 (ONLY FOR EMERGENCIES)

email: eleugeni@utmb.edu

Reviewer 2 Report

Dear authors, I am glad to read your work, which touches upon the topic of intercellular interaction, which is important for the scientific community, without which the existence of a multicellular organism is impossible. A detailed description of the discoveries that led to the emergence of a well-established theory of cell contacts and the possibility of transferring substances between different cells deserves close attention. The work consists of three massive blocks. The first describes gap junctions, the second describes the possibility of fusion of the membranes of two neighboring closely adjacent cells with the formation of pores. In the third, tunneling nanotubes and the role of mitochondrial transfer through them in the development of certain diseases. However, the connection between the three blocks is vague. There is no general idea that unites all three together. The existence of pores is shown only for a certain type of tissue, gap junctions are not able to pass the mitochondria. Thus, there is no generalization of information and a general conclusion from it. It may be worth restructuring the text, as well as adding a conclusion or conclusions or questions rises to emphasize the main idea and make it easier to understand.

In addition, it is required to expand the captions for Figures 4-6. Please, indicate from which articles they are taken. Give a diagram of the structure of the crayfish axon and indicate from which part of the body it was taken, for a better understanding of what is shown in the figure. In the caption to Figure 4, you need to add a description of the inset a - not only from whom the image was received, but also what exactly is depicted on it (it is better to indicate the most significant areas with arrows). Raise questions and some of the terms used in the signatures. For example, connective tissue. What is meant? It would be nice, what type of glia are on the sections?

And some small questions. Used different fonts in the titles. There is a missing paragraph on line 122. Needed to put this line in the title. Please explain what “current injection” on line 117 means. You need to decipher MV on line 177. What is the difference between non-excitable (line 107) and unexcitable (line 109)?

There is no certainty that it is permissible to use first-person narrative in reviews, even if it is data obtained personally by the authors of the review.

Author Response

May 23, 2022

To Dr. Megan Dong

Enclosed is our revised manuscript entitled "Direct cell-cell communication via membrane pores, gap junction channels and tunneling nanotubes: medical relevance of mitochondrial exchange " Manuscript number, IJMS-1729888. As requested by the reviewers and the Editors, we answered all the concerns and comments indicated by the reviewers. Thank you for the constructive comments and suggestions that certainly strengthened our manuscript. We included a point-by-point response to both reviewers, and the changes in the manuscript are noted in italics letters. We hope that our manuscript is now acceptable for publication in IJMS.

Reviewer #2.

Reviewer #2: Dear authors, I am glad to read your work, which touches upon the topic of intercellular interaction, which is important for the scientific community, without which the existence of a multicellular organism is impossible. A detailed description of the discoveries that led to the emergence of a well-established theory of cell contacts and the possibility of transferring substances between different cells deserves close attention. The work consists of three massive blocks. The first describes gap junctions, the second describes the possibility of fusion of the membranes of two neighboring closely adjacent cells with the formation of pores. In the third, tunneling nanotubes and the role of mitochondrial transfer through them in the development of certain diseases. However, the connection between the three blocks is vague. There is no general idea that unites all three together.

Answer: While it is true that gap junctional communication is quite different from that created by membrane pores and tunneling nanotubes, our goal was to cover all presently known aspects of direct cell-cell communication. We try to do better connections among the three different communication systems.

Reviewer #2: The existence of pores is shown only for a certain type of tissue, gap junctions are not able to pass the mitochondria. Thus, there is no generalization of information and a general conclusion from it. It may be worth restructuring the text, as well as adding a conclusion or conclusions or questions rises to emphasize the main idea and make it easier to understand.

Answer: Thank you for the comment; we included better conclusions in each section.

Reviewer #2: In addition, it is required to expand the captions for Figures 4-6. Please, indicate from which articles they are taken. Give a diagram of the structure of the crayfish axon and indicate from which part of the body it was taken, for a better understanding of what is shown in the figure.

Answer: The electron micrographs are all original unpublish images taken by one of the authors (C. Peracchia). All of the micrographs were taken from axons of crayfish abdominal ganglia.

Reviewer #2: In the caption to Figure 4, you need to add a description of the inset a - not only from whom the image was received, but also what exactly is depicted on it (it is better to indicate the most significant areas with arrows). Raise questions and some of the terms used in the signatures. For example, connective tissue. What is meant? It would be nice, what type of glia are on the sections?

Answer: We included the missing data and added additional information. A description was given: “Inset “a” shows a fluorescence micrograph of a median giant axon intracellularly injected with Lucifer Yellow CH; note that the nuclei of the adaxonal (sheath) glial cells are stained by the fluorescent dye” - not only from whom the image was received, but also what exactly is depicted on it (it is better to indicate the most significant areas with arrows). We have added red arrows pointing to the nuclei of the adaxonal (sheath) glial cells. Raise questions and some of the terms used in the signatures. For example, connective tissue. We changed it to “interstitial connective tissue.”

Reviewer #2: And some small questions. Used different fonts in the titles. There is a missing paragraph on line 122. No missing paragraph Needed to put this line in the title Done. Please explain what “current injection” on line 117 means Changed to “electrical current injection”. You need to decipher MV on line 177 Changed to “molecular weight (MW). What is the difference between non-excitable (line 107) and unexcitable (line 109)? Changed non-excitable to unexcitable

Answer: We added all the suggestions. Thank you.

Reviewer #2: There is no certainty that it is permissible to use first-person narrative in reviews, even if it is data obtained personally by the authors of the review.

Answer: Thank you. Changed to “the team of Dr. Terry A. Viancour”

Thank you for all your constructive comments and suggestions. We believe our findings are extremely novel and exciting and would appreciate their consideration for publication in IJMS. Changes in the text are indicated by red Italics letters.

Sincerely,

Eliseo Eugenin. Ph.D.

William D. Willis, Jr., M.D., Ph.D. Professorship in Neuroscience 

University of Texas Medical Branch (UTMB)

Department of Neuroscience, Cell Biology, and Anatomy 

105 11th Street 

Galveston, Texas, 77555

Office: (409) 772-7705

Lab    :(409) 772-0155

Cell Phone: (845) 490-3332 (ONLY FOR EMERGENCIES)

email: eleugeni@utmb.edu

Round 2

Reviewer 2 Report

Dear authors, thank you for your comments and explanations. However, I still have some doubts.

Connective tissue is composed of an extracellular matrix and several types of cells. The cells of connective tissue include fibroblasts, adipocytes, macrophages, mast cells and leucocytes. Accordingly, it is not correct to use this term in your work. The word "adjacent" may be used to describe this structure. As a last resort, you can use the expression used in your 1983 paper – «glial connective tissue sheath».

In your 1974 paper you use the term Schwann cell to refer to a particular type of glia. Why not use it now?

Did the axons of the crustaceans you examined have a myelin sheath and what does it look like under an electron microscope? Don't you think that what you call "connective tissue" is analogous to the myelin sheath?

By definition, exosomes are up to 100 nm in size, which raises doubts about their ability to carry mitochondria. Therefore, it is necessary to cite research references describing this process. Reference 112 does not appear in the text.

Author Response

To Dr. Megan Dong

Enclosed is our revised manuscript entitled "Direct cell-cell communication via membrane pores, gap junction channels and tunneling nanotubes: medical relevance of mitochondrial exchange " Manuscript number, IJMS-1729888. As requested by reviewer #2, we clarified some of clarified the EM data and added references about exosomes. We included a point-by-point response, and the changes in the manuscript are noted in italics red letters. We hope that our manuscript is now acceptable for publication in IJMS.

Reviewer #2.

Reviewer #2: Dear authors, thank you for your comments and explanations. However, I still have some doubts. Connective tissue is composed of an extracellular matrix and several types of cells. The connective tissue cells include fibroblasts, adipocytes, macrophages, mast cells and leucocytes. Accordingly, it is not correct to use this term in your work. The word "adjacent" may be used to describe this structure. As a last resort, you can use the expression used in your 1983 paper – «glial connective tissue sheath.»

In your 1974 paper, you use the term Schwann cell to refer to a particular type of glia. Why not use it now?

Answer: Thank you, we change it to Adjacent glial connective tissue sheath, non-cellular and glial satellite cell. Thus, we added the original names cited in the paper. Thank you.

Reviewer #2: Did the axons of the crustaceans you examined have a myelin sheath, and what does it look like under an electron microscope? Don't you think that what you call "connective tissue" is analogous to the myelin sheath?

Answer: Invertebrates lack myelin structures; however, we cannot discard that the connective tissue can help (and probably do) with the electrical transmission in a passive manner.

Reviewer #2: By definition, exosomes are up to 100 nm in size, which raises doubts about their ability to carry mitochondria. Therefore, it is necessary to cite research references describing this process. Reference 112 does not appear in the text.

Answer: We agreed and clarified that most manuscripts citing mitochondrial exchange focus on mitochondrial components, not the entire organelle. In addition, we reformat the references. Thank you.

Thank you for all your constructive comments and suggestions. We believe our findings are extremely novel and exciting and would appreciate their consideration for publication in IJMS. Changes in the text are indicated by red Italics letters.

Sincerely,

Eliseo Eugenin. Ph.D.

William D. Willis, Jr., M.D., Ph.D. Professorship in Neuroscience 

University of Texas Medical Branch (UTMB)

Department of Neuroscience, Cell Biology, and Anatomy 

105 11th Street 

Galveston, Texas, 77555

Office: (409) 772-7705

Lab    :(409) 772-0155

Cell Phone: (845) 490-3332 (ONLY FOR EMERGENCIES)

email: eleugeni@utmb.edu